# Molecular Mechanisms Regulating the DNA Repair Protein APE1: A Focus on Its Flexible N-Terminal Tail Domain

**DOI:** 10.3390/ijms22126308

**Published:** 2021-06-11

**Authors:** David J. López, José A. Rodríguez, Sonia Bañuelos

**Affiliations:** 1Biofisika Institute (UPV/EHU, CSIC) and Department of Biochemistry and Molecular Biology, University of the Basque Country (UPV/EHU), 48940 Leioa, Spain; david.lopezj@ehu.es; 2Department of Genetics, Physical Anthropology and Animal Physiology, University of the Basque Country (UPV/EHU), 48940 Leioa, Spain; josean.rodriguez@ehu.es

**Keywords:** apurinic/apyrimidinic endonuclease 1, APE1, DNA repair, BER pathway, abasic, nucleophosmin, NPM1, protein regulation

## Abstract

APE1 (DNA (apurinic/apyrimidinic site) endonuclease 1) is a key enzyme of one of the major DNA repair routes, the BER (base excision repair) pathway. APE1 fulfils additional functions, acting as a redox regulator of transcription factors and taking part in RNA metabolism. The mechanisms regulating APE1 are still being deciphered. Structurally, human APE1 consists of a well-characterized globular catalytic domain responsible for its endonuclease activity, preceded by a conformationally flexible N-terminal extension, acquired along evolution. This N-terminal tail appears to play a prominent role in the modulation of APE1 and probably in BER coordination. Thus, it is primarily involved in mediating APE1 localization, post-translational modifications, and protein–protein interactions, with all three factors jointly contributing to regulate the enzyme. In this review, recent insights on the regulatory role of the N-terminal region in several aspects of APE1 function are covered. In particular, interaction of this region with nucleophosmin (NPM1) might modulate certain APE1 activities, representing a paradigmatic example of the interconnection between various regulatory factors.

## 1. Introduction

Our genome is constantly exposed to both endogenous, i.e., of metabolic origin, and exogenous sources of DNA damage. If left unrepaired, damage to DNA is highly deleterious and may lead to diseases such as cancer. To protect the genetic material, cells have evolved a number of complex machineries, able to sense and repair DNA lesions or, when the damage load is too high, to arrest cell growth and/or induce apoptosis [1]. The different cellular processes triggered by DNA lesions are globally coordinated in the so-called DNA damage response (DDR) [2]. One of the main DDR machineries is the base excision repair (BER) pathway [3,4]. This pathway is essential in managing one of the most frequent types of DNA damage, i.e., mostly oxidative and alkylative, non-bulky base lesions. A key enzyme in the BER pathway is APE1 (DNA (apurinic/apyrimidinic site) endonuclease 1) [5].

Human APE1 (also known as APEX nuclease, HAP-1, APEX1, APEN, redox factor 1, and REF-1) is a ubiquitously expressed 318 amino acid long protein with multiple functions. The main enzymatic activity of APE1 is the incision of abasic sites in DNA [6]. As a central enzyme in BER, APE1 is an essential protein in mammals since early embryonic development [7]. Altered expression levels and/or subcellular distribution of APE1 have been reported in several types of tumours [8], and APE1 is considered a relevant target in cancer therapy. In particular, pharmacological inhibition of APE1 may enhance tumour sensitivity to DNA damaging agents [8,9].

In addition to acting as a DNA repair protein, APE1 also fulfils additional functions. In fact, it was independently discovered as a redox factor regulating the DNA binding of several transcription factors involved in different growth signalling pathways [10] and was subsequently found to also be involved in additional genetic regulatory mechanisms [11,12,13,14]. For instance, it acts as a transcriptional repressor through interaction with negative calcium-response elements (nCaRE) [12]. Interestingly, novel functions of APE1 in the RNA world are becoming increasingly evident. Thus, it has a putative “quality control” role for RNA, which is more prone to oxidative damage than DNA [15].

The mechanistic bases of APE1 activity in DNA repair have been deeply characterized at both biochemical and structural levels. In particular, crystallographic data on APE1 complexes with substrate and product-mimicking oligonucleotides have untangled the details of APE1 catalytic activity [16,17]. However, the regulatory mechanisms that modulate APE1 function are less understood. Similar to many other proteins, regulation of APE1 operates mainly through changes in subcellular localization, post-translational modifications (PTMs), and interactions with other proteins. By being primarily involved in these three factors, the flexible N-terminal segment of APE1 (residues 1–61) undoubtedly plays a prominent role in regulating its activity.

Here, we review the various cellular functions of APE1, discussing the possible regulatory operation of its N-terminal tail. This evolutionarily acquired region, typical of eukaryotic APE1 homologues, may hold the clue to how the multifaceted functions are fine-tuned in this enzyme.

## 2. The Role of APE1 in the BER Pathway

The BER pathway has evolved to deal with damaged nitrogenous bases, abasic sites, and several types of single strand breaks (SSBs) [4,18]. BER predominantly manages small, non-helix distorting nucleobase lesions. Other non-bulky lesions such as interstrand crosslinks [19] and cyclopurines [20] are instead usually corrected by alternative pathways. In the context of BER, altered bases are first detected and removed by lesion-specific DNA glycosylases, usually resulting in an abasic (apurinic/apyrimidinic, AP) site (Figure 1). AP sites can be also generated through spontaneous or damage-induced depurination (more frequent than depyrimidation), caused, for example, by reactive oxygen species (ROS). It has been estimated that more than 10,000 AP sites, which can arrest both DNA replication and transcription, are normally generated per day in our cells [21]. Since they are non-informative, AP sites may be mutagenic, and can promote highly deleterious interstrand crosslinks [22].

AP sites are recognized by APE1, which catalyzes the incision of the phosphodiester bond in their 5′ side, leaving a break with 3′-hydroxyl and 5′-deoxyribose phosphate (5′-dRP) termini (Figure 2A). Of note, some BER-initiating DNA glycosylases, termed “bi-functional”, such as OGG1, which excises 8-oxoguanine, are also capable of cleaving the DNA strand at the 3′ side of the lesion, thus, rendering a 3′ α,β-unsaturated aldehyde (PUA) [4]. In addition to its endonuclease activity, APE1 also displays a 3′ to 5′ exonuclease activity that subsequently removes the terminal PUA group [23]. Thus, APE1 involvement is manifold and warranted in most BER subpathways, displaying either its endonuclease role, or 3′ exonuclease activity, for the cleansing of blocking non-productive terminal groups generated in BER intermediates [4,5]. A further subgroup of bi-functional glycosylases (e.g., NEIL1 and NEIL2) cuts the DNA strand through β,δ-elimination, leaving a 3′ phosphate, which is removed by polynucleotide kinase (PNK) [4].

After DNA strand incision by APE1, downstream BER enzymes, usually DNA polymerase β (Pol β) and DNA ligase III, further coordinated by the scaffolding protein XRCC1 (X-ray repair cross-complementing protein 1), complete the base correction, i.e., Pol β, with its lyase domain, excises the 5′-dRP group left behind by APE1, and simultaneously fills the gap with its polymerase activity [24]. The nick is then sealed by DNA ligase III. This pathway, depicted in Figure 1, is called “short-patch BER” (SP-BER), or “single-nucleotide BER” (SN-BER). A variation of BER, termed “long-patch” (LP-BER), is thought to be used in the case of clustered oxidative lesions. In this alternative pathway, polymerases δ and/or ε add 2–10 nucleotides with strand displacement, and the DNAse FEN1, in coordination with PCNA, removes the resulting flap prior to ligation by ligase I (Figure 1) [25]. Thus, LP-BER seems to utilise DNA replication proteins and, indeed, is thought to be preferred over SP-BER during S phase [26]. The choice of the short- or long-patch BER pathway depends on multiple factors such as initiating glycosylase, cell type, cell cycle phase, availability of BER enzymes, and regulatory proteins [27].

Coordination of BER relies on the scaffolding protein XRCC1, which rapidly accumulates at sites of DNA damage. XRCC1 interacts with, stabilizes, and stimulates the activity of various BER enzymes, including APE1 [28]. Prior to XRCC1 accumulation on DNA lesions, these sites (nicks, gaps, etc.) are thought to be detected and bound by poly (ADP-ribose) polymerase 1 (PARP-1), which post-translationally modifies proteins (including itself) through addition of poly(ADP-ribose) (PAR) units (a process termed PARylation). PARP-1 PARylation promotes the recruitment of XRCC1, APE1, Pol β, and DNA ligase III to the lesion site [4]. Thus, the BER machinery is thought to operate on chromatin-associated platforms where the various enzymes of the complex handle the DNA in a concerted fashion, so-called “passing-the-baton” [29], ensuring optimal efficiency. Substrate channelling between BER enzymes has been demonstrated in vitro [29]. In the particular case of APE1, rapid catalysis is followed by slow product release in vitro [30], while turnover seems to be enhanced by Pol β. This suggests that the intrinsic slow dissociation of the product may help to conceal potentially cytotoxic incision intermediates until the downstream enzyme is available [4,18,28].

While the major and best characterized activity of APE1 in BER is as an endonuclease, this enzyme contributes additional roles in the context of DNA repair. These other functions include removal of terminal blocking groups in BER intermediates and proofreading DNA mismatches introduced by Pol β, making use of its 3′ end processing activities [23]. Furthermore, in addition to its roles in BER, APE1 has been shown to participate in a DDR subpathway known as nucleotide incision repair (NIR), by incising non-abasic DNA at certain base lesions [23]. Moreover, APE1 also contributes to the repair of SSBs [23,31], by sensing these lesions and initiating their 3′–5′ resection, thanks to its exonuclease activity [32]. This role of APE1 in SSB repair is then continued by the endonuclease APE2, which promotes the ATR-Chk1 pathway.

## 3. Roles of APE1 beyond DNA Repair

Beyond its DNA repair-related activities described above, APE1 has been shown to also play roles in redox-related cellular functions, RNA metabolism, and DNA regulatory processes. While this review mostly focuses on how the DNA repair functions of APE1 are regulated, non-DNA repair activities of APE1 are briefly described in this section, as they are certainly important for cell function, and may be critical in specific cellular contexts.

APE1 has a long known redox-related function and, in fact, one of APE1 aliases is “redox effector factor 1” (REF1). On the one hand, APE1 regulates the DNA binding activity of important transcription factors such as AP-1, p53, HIF1α, and NF-κB by maintaining them in a Cys-reduced, active state [10,33]. Thus, APE1 provides a redox-dependent mechanism for the regulation of target gene expression. From a mechanistic standpoint, it has been suggested that a disulfide bond forms between Cys residues of the N-terminal half of the protein [34], but the details of APE1 redox function still remain largely unclear. Interestingly, crosstalk between the repair and redox activities of APE1 may take place in the context of the response to oxidative stress, as exemplified by the regulation of VEGF upon hypoxia [35], and by the IgA class switch recombination [36].

On the other hand, similar to several other DNA repair proteins, APE1 has been implicated in RNA-related processes [15]. APE1 dynamically interacts with many proteins involved in RNA processing [15] and is able to bind more than 1000 RNA transcripts, according to a RIP-seq analysis [37]. Binding of APE1 to structured RNA molecules involves its N-terminal 33 amino acids [38]. APE1 possesses 3′-RNA phosphatase and 3′ exoribonuclease activity [39] and it is able to cleave abasic single-stranded RNA through its endonuclease activity [40]. Probably in coordination with nucleophosmin (NPM1), APE1 has been related to a putative “quality control” mechanism of ribosomal RNA (rRNA) in the nucleolus (see below) [15,37,41]. In this regard, APE1 binds several rRNA species in vivo, regulates rRNA oxidation levels, and its deficiency leads to increased RNA oxidation [42]. APE1 function in RNA metabolism affects gene expression and may have an impact on cancer development. A clear example is direct cleavage of c-myc mRNA by APE1, which seems to play a role in regulating the expression of this proto-oncogene [43]. A second example relates to the proposed role of APE1 in microRNA (miRNA) biogenesis [44]. In tumour samples, APE1 has been shown to enhance the post-transcriptional maturation of oncogenic miRNAs that modulate PTEN expression, thus, leading to lower levels of this tumour suppressor. These findings provide novel bases for the use of APE1 as a target in cancer therapy.

Finally, novel functional implications of APE1 in DNA regulatory processes have been unveiled recently. On the one hand, APE1 participates in the BER pathway-mediated correction of mutagenic T:G mismatches generated when 5-methylcytosine, the key DNA epigenetic mark, is spontaneously deaminated, becoming a thymine [13]. On the other hand, APE1 has been recently shown to regulate the formation of G-quadruplex (G4) structures in guanine-rich DNA sequences [14]. Guanines are highly susceptible to oxidation, forming 8-oxoguanine (8-oxoG), a prevalent base lesion that is repaired by the BER pathway involving the activity of APE1. Adoption of G4 structures has emerged as a structure-based epigenetic regulatory mechanism with implications in the control of transcription, replication, and telomere maintenance [45]. In this regard, binding of APE1 to damaged G-rich sequences in gene promoters has been shown to promote G4 formation and facilitate the loading of transcription factors to regulate gene expression [14].

## 4. APE1 Structure

Human APE1 mainly consists of a globular domain, where the nuclease activity resides, preceded by an N-terminal, ca. 40 residues long, intrinsically disordered region [46]. Crystallographic studies have established that the nuclease domain is shared by the phosphodiesterase superfamily of enzymes, with a common four-layered α/β-sandwich fold (Figure 2B), where two symmetrically related similar halves can be distinguished [17,47]. APE1 is homologous to other nucleases, such as DNAse I and *E. coli* AP endonuclease EXOIII. A structural comparison of the three enzymes revealed loop regions specific to APE1 and EXOIII that explain their specificity for abasic residues [47].

For catalysis, APE1 kinks the DNA helix by inserting loops within both the major and minor grooves, to recognize the deoxyribose moiety in an extra-helical (“flipped-out” or “everted”) conformation [17,48] (Figure 2B). This sculpting of the DNA provides specificity for the damage, since undamaged DNA cannot be similarly contorted [48]. High resolution crystal structures of APE1 bound to substrate and product-mimicking oligonucleotides that include tetrahydrofuran (THF) as a stable AP site analogue have provided snapshots of the pre- and post-incision complexes [16]. Cleavage is afforded by the nucleophilic attack of a highly coordinated (by Asp210, Asn212) water molecule on the 5′ phosphorous. One Mg^2+^ atom, being coordinated by Asp70, Glu96, and a water molecule, is needed for the catalysis, as it helps to stabilize a reaction intermediate.

**Figure 2 ijms-22-06308-f002:**
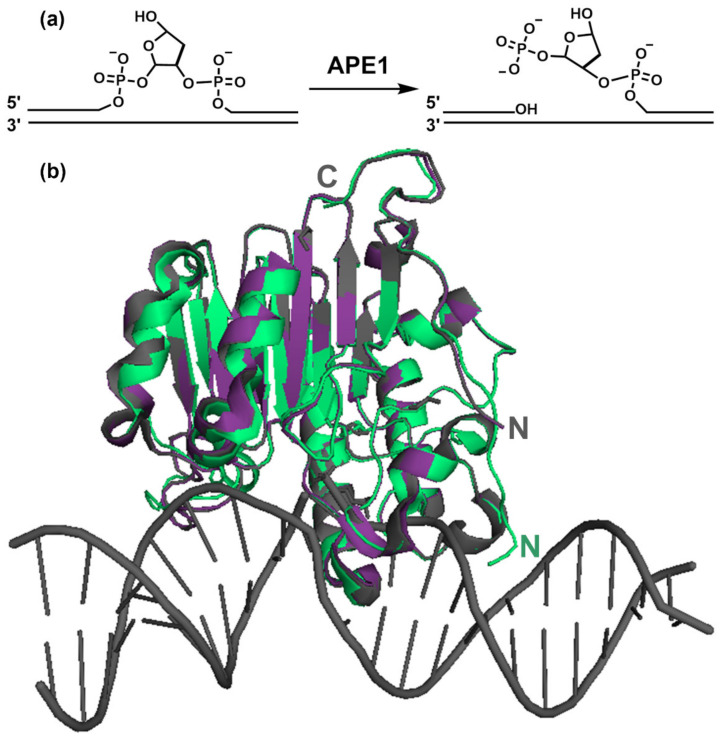
(**a**) Incision reaction catalyzed by APE1; (**b**) APE1 3D structure. Structural models of human APE1 either free, in green (accession code 4QHE) [49], or bound to abasic substrate and product-mimicking DNA oligonucleotides (magenta, 5DFI and grey, 5DFF, respectively [16]). The DNA structure corresponds to entry 5DFF. The structures of the substrate and product complexes have a root mean square deviation (r.m.s.d.) of 0.62 Å between them (referred to C_α_ atoms) and 1.19 Å and 1.36 Å as compared with free APE1, respectively. “N” and “C” indicate the N- and C- termini, respectively. The models start with residue 38 (4QHE) or 43 (5DFI and 5DFF), and thus do not provide information on the N-terminal tail of APE1. The figure was prepared with PyMol [50].

A flurry of structures of APE1 bound to different substrates has further elucidated the molecular basis of the additional capabilities of the enzyme, such as its 3′ exonuclease activity and the recognition of base mismatches [23,51,52]. The structural information available generally reflects that the APE1 active site is quite rigid [17] and that catalytic differences between substrates mainly result from altered DNA conformations [52]. This rigidity is a feature typical of DNA-contorting enzymes [48].

Crystallographic data have been supported by biochemical characterization of the catalysis, providing a complete picture of the APE1 enzymatic mechanism. Remarkably, the 3D structure of the globular nuclease domain does not significantly change when APE1 binds to DNA [17], regardless of the DNA molecule being the abasic substrate or product [5,16] (Figure 2). Nevertheless, by using FT-IR spectroscopy, we have observed that binding to DNA does have an influence on the secondary structure and thermal stability of full-length APE1 in solution [53], suggesting that the conformation of the protein in solution somehow senses the interaction.

Importantly, the 3D structures do not provide information on an important region of the protein, i.e., its N-terminal tail. In the available APE1 structural models, the segment spanning at least 37 N-terminal residues is either absent in the APE1-expressing construct used [16,47] or not visible in the structure due to its flexibility [17]. Information on the N-terminal segment is also lacking in 3D models of APE1 in complex with DNA [16,17], even if the highly basic character of this region suggests that it contributes to the interaction with the DNA. In particular, the abundant Lys residues of the first half of the N-terminal region (Figure 3) might establish contacts with the DNA backbone phosphates in the recognition event. Interestingly, a higher protection pattern of lysines 24–35 towards ^1^H/^2^H exchange has been observed by mass spectrometry in an intact APE1–abasic DNA duplex complex as compared with the post-incision complex [54], reflecting that this region may contribute to distinguish between DNA substrate and product.

Although the structural role of the N-terminal tail of APE1 is still to be established, mounting evidence points to this region as a key feature in APE1 activity and regulation [13,26,55,56].

## 5. Role of the N-Terminal Tail in APE1 DNA Repair Activity

Flexible N- or C-terminal regions are common in DNA binding proteins, particularly in BER enzymes, where they play a critical regulatory role [26]. Some of these regions are only present in the protein homologues of higher organisms. For example, the disordered N- or C-terminal tails of DNA glycosylases are characteristic of mammalian enzymes [26], where they exhibit a remarkably high variability in sequence and length. Similarly, the N-terminal 61 residues of human APE1 (Figure 3) are absent from prokaryotic AP endonucleases, such as Xth from *E. coli* [26,56]. The N-terminal tail of APE1 exhibits higher sequence variability than the globular domain. However, it is worth noting that the first 35 amino acids within APE1 N-terminal tail (a highly basic segment, rich in Lys residues, with isoelectric point 9.7) are specifically conserved in mammals [56].

The N-terminal 61 residues of APE1 are clearly important for the redox function of the enzyme. Thus, APE1 redox activity has been mapped to the first 127 residues, possibly involving Cys65 (only present in mammalian homologues) and Cys93. Moreover, deletion of the N-terminal 61 residues results in a significant loss of stimulation of DNA binding by the transcription factor AP-1 [57]. In contrast, being absent in prokaryotic APE1 homologues [26], and dispensable for its endonuclease activity [46], the contribution of the N-terminal region to APE1 repair functions in the BER pathway is less evident.

As mentioned above, although most of the APE1 N-terminal tail is missing from the 3D structures, it probably contributes to APE1/DNA interaction. In fact, the N-terminal tail does affect APE1 binding to DNA, as well as the incision catalytic properties [42]. In addition, it could be important in scanning the DNA for the presence of lesions, in coordinating APE1 with other BER enzymes, or in the BER/NIR dichotomy.

**Figure 3 ijms-22-06308-f003:**
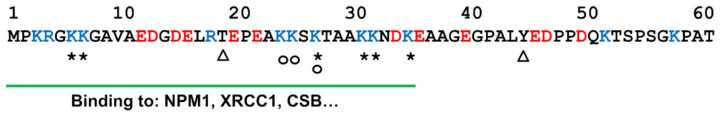
Amino acid sequence of human APE1 N-terminal region. The sequence corresponds to Swiss-Prot entry P27695-1. Positively and negatively charged residues are highlighted in blue and red, respectively. Acetylation (*), phosphorylation (∆), and ubiquitination (o) sites are indicated, as well as the binding region for NPM1 [42], XRCC1 [28], and CSB (Cockayne syndrome B) [58].

APE1 recognition and catalysis properties are additionally modulated by DNA conformational properties. In this regard, APE1 has been described to preferentially recognize AP lesion sites in double-stranded DNA [17,40,47]. Nevertheless, APE1 is also active (conceivably with different binding and/or catalytic properties) on AP sites within alternative contexts, such as single-stranded DNA or G-quadruplexes. The remarkable variety of APE1 substrates (i.e., AP sites and also DNA-blocking termini and base mismatches) suggests that the enzyme may detect conformational distortions along the DNA helix. Considering the rigid nature of the active site [17,23], these observations invoke a role for the flexible N-terminal tail in the different recognition events. This is most relevant, as APE1 has to quickly scan the chromatin in search for AP sites and other defects.

Similar to what has been described for other DNA binding proteins, the N-terminal region of APE1, given its length, conformational flexibility, and basic character, could be involved in the scanning of the chromatin seeking abasic sites or other lesions. A critical step for early BER proteins (DNA glycosylases and APE1) is thought to be the recognition of non-bulky lesions that do not significantly distort the DNA helix [26]. Many DNA binding proteins exhibit disordered segments that are in charge of scanning in search for the specific binding sites. Such scanning is commonly mediated by non-specific, mostly electrostatic, transient interactions, between these tails, often of basic character, and the DNA [26,59]. Indeed, a “monkey bar” mechanism has been hypothesized, whereby the flexible tails help those proteins to slide and also jump between DNA segments [59].

The N-terminal tail also contributes to coordinate APE1 with the rest of enzymes of the BER machinery in the context of multi-protein BER complexes [28]. Within these complexes, probably organized by the scaffolding protein XRCC1, each enzyme hands its product to the next one, following a so-called “passing the baton” model [4,28,60]. In this model, APE1 has been described to stimulate product release by DNA glycosylases and, in turn, APE1 turnover is facilitated by Pol β and by XRCC1. In both cases, APE1 N-terminal tail plays an important role [28,61,62].

A prominent question regarding the regulation of APE1 function in BER is how Pol β and/or XRCC1 enhance APE1 catalytic turnover. As mentioned above, release of the incised DNA product is the rate-limiting step in the activity of APE1 [30]. A truncated APE1 lacking the N-terminal 33 residues (APE1 ∆N33) is slightly more active than wild type APE1 due to a decreased affinity for the incised DNA, suggesting that the N-terminal segment is responsible for retaining APE1 on the reaction product [55,63]. This retention needs to be relieved when the next enzyme in the BER route, Pol β, is ready to accept the intermediate. Through binding to the N-terminal tail of APE1, XRCC1 seems to help the enzyme detach from the DNA to promote turnover [28]. Therefore, the N-terminal tail of APE1 appears to help keep the enzyme bound to the product DNA until it is timely handed off to the next enzyme of the route in a coordinated manner. While not essential for binding and endonuclease activity of APE1 on double stranded DNA, the N-terminal tail is apparently required for APE1 binding to ssDNA and RNA. In these cases, in contrast to dsDNA, no product inhibition effect on the hydrolysis rate has been observed [55].

Finally, the N-terminal tail of APE1 seems to have a particularly prominent role in the context of the NIR pathway, since a truncated variant lacking the first 61 residues is proficient in BER but exhibits a significantly decreased NIR activity [64]. Therefore, that region, together with other factors, (e.g., pH and divalent cations), may contribute to modulate APE1 substrate specificity, and thus BER/NIR dichotomy [64].

## 6. Role of the N-Terminal Tail in Regulating APE1 Localization, Post-Translational Modification, and Protein–Protein Interactions

In addition to the direct contribution of the N-terminal tail to APE1 repair activities described above, this domain is important to finely tune APE1 function by additional mechanisms. Similar to most proteins, APE1 functionality is modulated by subcellular localization, post-translational modifications (PTMs), and interaction with other proteins (Figure 4). In this regard, the conformational flexibility of the N-terminal tail is ideally suited for harbouring cell localization signals, being the subject of PTMs and establishing interactions with other proteins. A crosstalk between these three factors further regulates APE1, which is described in detail below. For example, acetylation of residues in the N-terminal tail affects the subcellular localization of the protein, and also binding to NPM1. This interaction, in turn, determines APE1 enrichment in the nucleolus.

### 6.1. The N-Terminal Tail and APE1 Subcellular Localization

While being mainly located within the cell nucleus, and particularly enriched in nucleoli [38], APE1 is a nucleo-cytoplasmic shuttling protein that, depending on the physiological status of the cell, can also be found in other cellular compartments [65,66], or even in the extracellular milieu [67]. APE1 subcellular traffic is mediated by different localization signals, and further regulated by protein–protein interactions and by PTMs. Shuttling of APE1 between the nucleus and cytoplasm relies on the nuclear transport receptors importin α/β and CRM1 (exportin 1) [68,69]. The first seven residues of APE1 harbour a classical nuclear localization signal (NLS), with residues Glu12 and Asp13 also playing a role in import [68]. Further supporting the role of the N-terminus in APE1 nuclear import, a naturally occurring form of APE1 lacking the first 31 amino acids (the product of granzyme A-induced proteolysis) has been detected in the cytoplasm, endoplasmic reticulum, and mitochondria, especially in cells committed to apoptosis [70]. The cleavage of APE1 may prevent the cell from repairing damage and recovering, thus, enhancing granzyme-mediated cell death [70].

Nuclear export of APE1 is positively regulated by MDM2-mediated ubiquitination [71] and is also stimulated in the presence of nitric oxide (NO). In this latter case, the mechanism may involve S-nitrosylation of Cys 65, 93 and/or 310, perhaps leading to the exposure of a predicted CRM1-dependent NES in amino acids 64–80 [69], whose activity remains to be experimentally validated.

Moreover, translocation of APE1 from the cytoplasm to the mitochondria, where it is thought to be involved mtDNA repair [65,72], is mediated by ROS signalling and Arg301 methylation and depends on Tom20 [73]. Once in the intermembrane space, which is thought to represent a storage site, a rise of mitochondrial ROS triggers a rapid translocation of APE1 into the matrix, through the Tim23 complex [74]. A putative mitochondrial targeting sequence (MTS) has been identified in the C-terminus of the protein and it has been proposed to be normally masked by the N-terminal tail in the intact protein [63].

The relative enrichment of APE1 in the nucleolus is probably determined by its interaction with the nucleolar protein nucleophosmin (NPM1) [42] and requires active rRNA transcription. As described below, interaction with NPM1 is mediated by APE1 N-terminal tail. The nucleolar localization of APE1 may dynamically respond to different stimuli. Thus, oxidative stress has been described to induce its release from the nucleolus and its subsequent recruitment to “patches” on the chromatin [75], a relocation that seems to be paralleled by NPM1 [53]. Similarly, genotoxic and oxidative stress, as well as UVA irradiation, promote the enrichment of APE1 in nuclear speckles, co-localized with OGG1 [76].

Finally, recent studies have shown that APE1 can be secreted to the extracellular milieu, and that serum APE1 may represent a novel prognostic biomarker in certain types of cancers. In fact, its concentration in the serum of hepatocellular carcinoma and non-small cell lung cancer patients is increased as compared with healthy controls [67,77]. APE1 has been found to be secreted in active form and through exosomes in response to genotoxic treatments [78].

### 6.2. The N-Terminal Tail and APE1 Post-Translational Modifications

The function of APE1 is modulated through several PTMs. As other BER enzymes, such as XRCC1 and Pol β, APE1 has been described to be modified by PARP1-mediated PARylation [79] which, as described for PARP1 itself [80], probably influences the DNA scanning rate of APE1. In addition, PARP1 is able to stimulate DNA incision by APE1 [29]. Moreover, phosphorylation of Thr233 regulates APE1 redox activity [81], and S-nitrosylation of three cysteines (see above) affects its nuclear export [69]. Recently, a novel PTM of APE1 has been reported, i.e., methylation of arginine residue Arg301, which promotes translocation of the protein to the mitochondria [73].

PTMs often take place on flexible protein regions [82] and, indeed, multiple residues in the N-terminal tail of APE1 undergo modifications, such as phosphorylation, acetylation, or ubiquitination (Figure 3). Phosphorylation of Thr19 and Tyr45 has been detected in proteomic surveys (PhosphoSite Plus database [83]). In addition, acetylation of Lys residues 6 and 7 by CBP/p300 affects APE1 transcriptional regulatory functions [12,84], as well as the performance of the NLS [68]. Acetylation of Lys residues 6 and 7 is induced upon genotoxic stress and is antagonized by sirtuin-1 (SIRT1) deacetylase [85]. In addition, APE1 lysines 27, 31, 32, and 35 also undergo acetylation, which has been related with a possible inhibition of APE1 binding to RNA and to NPM1 [55]. In this regard, acetylation might be seen as abrogating the role of the N-terminal tail, but it must be noted that acetylated APE1 does not behave exactly as a truncated version of the protein lacking the first 33 aminoacids (∆N33) [55]. Given that the scanning ability of APE1 along the DNA is probably governed by electrostatic interactions, acetylation of N-terminal tail residues would most likely have an impact on this activity.

Finally, MDM2-mediated ubiquitination of Lys residues 24, 25, and 27 causes exclusion of APE1 from the nucleus and targets it for degradation, when DNA repair is abandoned in apoptosis [71,84].

### 6.3. The N-Terminal Tail and APE1 Protein–Protein Interactions

As discussed above, interaction of APE1 with other proteins within the BER multi-protein complexes assembled on the chromatin [28] is crucial for the coordination of the pathway, but APE1 interacts with many other cellular proteins. In this regard, the BioGRID database [86] reports 1024 unique interactors for APE1. These include, among others, protein ligands responsible for the subcellular trafficking of the protein, the enzymes mediating its post-translational modification (acetyl transferases, kinases, etc.), and transcription factors that are the subject of APE1 redox activity [44].

Given the advantages of intrinsically disordered segments for establishing molecular interactions [87], it is not surprising that the N-terminal tail of APE1 mediates many of its best-characterized interactions, including binding to XRCC1 and NPM1 [28,42]. In particular, the N-terminal 33 residues of APE1 constitute the binding site for NPM1 [42], which, as detailed below, plays a major role as a regulator of APE1.

## 7. Nucleophosmin as a Regulator of APE1 via the N-Terminal Tail

NPM1 is an abundant nucleolar protein that performs several functions affecting cell homeostasis. It is an oligomeric, multi-domain protein of 294 residues, consisting of a pentameric core [88] connected, through very long, flexible linkers, to small, globular C-terminal domains [89]. NPM1 is involved in ribosome assembly and export, control of centrosome duplication, stabilization of tumour suppressors (e.g., p53 and Arf), and plays a chaperone role in the nucleolus under stress conditions [90,91]. Recent reports have indicated that NPM1 might take part in different DDR pathways [92], such as the repair of DSBs [93] and “translesion synthesis” (TLS) repair [94]. Although normally enriched in the nucleolus, NPM1 shuttles between nucleoli, nucleoplasm, and cytoplasm. It interacts with nucleic acids, as well as with many proteins, and is considered a “hub” of the nucleolar interactome [91].

NPM1 was first identified as an APE1 interactor on a proteomics survey and has been shown to be responsible for APE1 accumulation in the nucleolus [42]. The protein domains mediating APE1/NPM1 interaction have been characterized [42,53]. Upon oxidative stress, both APE1 [75] and NPM1 [53] are partly released from nucleoli, and subsequently recruited to chromatin regions that could correspond to BER repair platforms. Importantly, NPM1 stimulates incision of abasic DNA by APE1 in vitro [42,53]. This stimulatory effect could be explained by the fact that NPM1 favours specific binding of APE1 to abasic DNA, while competing with off-target associations [53]. Considered together, these results suggest that APE1 and NPM1 might cooperate in the BER pathway (Figure 5): Through binding to the N-terminal tail of APE1, NPM1 would keep the enzyme in an open conformation, ready for selective and more efficient binding to abasic sites. Once the incision has taken place, NPM1 could facilitate APE1 release from the product [53]. Additionally, the protein chaperone ability of NPM1, and the fact that it can bind several molecules of APE1, suggest that it could act as a reservoir of APE1 or facilitate its transport to lesion sites on the DNA.

When both proteins reside in the nucleolus, NPM1 could mediate the reported role of APE1 in the context of RNA quality control [95]. While NPM1 stimulates APE1 endonuclease activity on double-stranded abasic DNA, it has been described to inhibit endoribonuclease activity on single-stranded RNA containing AP sites [55]. The activity of APE1 and NPM1, in the context of RNA metabolism, seems to reflect the more general role of the nucleolus as organizer of stress response and might take place mostly in the absence of DNA damage [96]. Upon genotoxic stress, acetylation of Lys residues in the N-terminal tail of APE1 leads to both dissociation from NPM1 and nucleolar exit of APE1 [38], perhaps to reinforce its BER repair activity.

Once outside the nucleolus, binding of NPM1 to the N-terminal tail of APE1 could also modulate the DNA scanning properties of APE1, and thus lesion detection and BER efficiency. In fact, APE1 has been reported to form filaments on DNA fragments in vitro [97], a phenomenon that could be related to its DNA scanning ability. A truncated ∆N61 APE1 mutant multimerizes on DNA much less efficiently [62], suggesting that the N-terminal tail has a role in this mechanism.

NPM1 is considered to be an oncoprotein and constitutes a therapeutic target in cancer [98]. In particular, NPM1 is the most frequently mutated protein in acute myeloid leukaemia (AML), accounting for approximately one third of the patients [99]. AML-linked NPM1 mutants are aberrantly located to the cytoplasm of tumour cells, and seem to induce mislocalization of some ligands, such as APE1. This suggests that BER repair could be defective in those tumour cells [100], a feature that might be related to the onset of the disease, but also represents a potential weakness to be exploited for therapeutic intervention.

## 8. APE1 as a Therapeutic Target

As the primary mechanism for repairing oxidative damage, the BER pathway is essential for maintaining the genome integrity, and thus its imbalance can increase the risk of cancer and neurodegenerative diseases [4]. Alterations in the expression level and subcellular distribution of APE1 have been described in several types of tumours [8], cytoplasmic staining possibly reflecting its role in mitochondrial DNA repair under high ROS load [101]. Moreover, since cancer chemotherapy and radiotherapy often inflict damage to the DNA in the form of AP sites, base alkylation, SSBs, etc., the BER capacity is an important factor determining clinical response to treatments [102]. Of note, the occurrence of clustered lesions [19] is probably significant in this scenario, and may affect the proficiency of BER, for example, inhibiting APE1 in some conditions [103]. Thus, overexpression of APE1 is found in some types of tumours and is associated with chemoresistance [104]. Indeed, targeting BER is a currently proposed strategy to sensitize tumour cells towards treatments [8,9,104]. Several APE1 inhibitors (of the repair activity, redox activity, or both) have been identified by high throughput screening of chemical libraries, able to potentiate the cytotoxic effect of alkylating agents [8]. The coordination with other proteins could be an alternative approach for targeting APE1. In this context, Poletto et al. found a correlation between APE1 and NPM1 expression levels in ovarian cancer and explored the novel strategy of targeting their interaction [105].

## 9. Conclusions and Open Questions

The N-terminal tail of APE1 has crucial roles in the regulation of its function and in BER coordination. In the context of BER, this domain probably helps the enzyme to scan the DNA in search of lesions, and to remain attached to the incised product until release is permitted by the subsequent enzymes. APE1 catalytic turnover is facilitated by Pol β, XRCC1, and probably NPM1. This modulation, at least in the case of XRCC1 and NPM1, is mediated by binding to APE1 N-terminal tail, but the mechanistic details remain to be fully elucidated.

BER regulation is probably more complex than anticipated, the particular subpathways depending, for example, on the cell cycle phase, damage burden, and the DNA sequence context of the lesion [26]. We must learn how these different situations are signalled for APE1 modulation, and to what extent they operate via the N-terminal tail.

The flexible character of APE1 N-terminal tail hampers elucidation of its structural features. Remarkably, although the N-terminal tail is probably involved in the interaction, binding of APE1 to nucleic acids is not enough to fix the structure of this region and solve its position in crystals made with the full-length protein [17]. This suggests that, even in this setting, the N-terminal tail is mobile and establishes, at best, transient interactions that can be dynamic and change along the catalysis process. Indirect methods, such as spectroscopy, may be of help to explore the structural role of this region. In our hands, circular dichroism and infrared spectroscopy were sensitive to conformational changes in APE1 upon DNA binding [53]. Yu et al. could capture “shadows” of APE1, including specific residues of the N-terminal tail, when bound to a DNA substrate or product, based on proton/deuterium protection patterns, and they reported interesting differences in that region [54].

APE1 regulatory mechanisms probably imply dynamic variations in the relative positions of the N-terminal tail with respect to the globular domain and of both domains with respect to the DNA. Another methodological approach to evaluate intra- and intermolecular conformational rearrangements that may underpin APE1 function and regulation is Förster resonance energy transfer (FRET), a fluorescence spectroscopy technique sensitive to the distances between two fluorophores (donor and acceptor of the transference) [106]. In particular, novel single-molecule methodologies provide us with the spatial and temporal resolution to follow molecular movements that could be hidden in the average of sample populations [107] and can be applied to proteins containing intrinsically disordered regions [108], such as the N-terminal tail of APE1.

Finally, the fact that the N-terminal tail has been evolutionary acquired suggests that it is necessary for APE1 functions different from the ancestral endodeoxyribonuclease found in prokaryotes. These functions, which could include the redox activity, the participation in RNA metabolism, and epigenomic regulation to name a few, seem to imply a more sophisticated regulatory level, typical of higher eukaryotes. For example, the N-terminal tail seems to “encumber” APE1 in the catalysis on DNA, while it may be necessary for the RNA functions of APE1 in the nucleolus [95]. In this context, it would be interesting to explore whether different subdomains within the N-terminal tail (e.g., the most N-terminal part, more conserved in mammals, and the distal part, involved in the redox activity) may have specific roles.

## Figures and Tables

**Figure 1 ijms-22-06308-f001:**
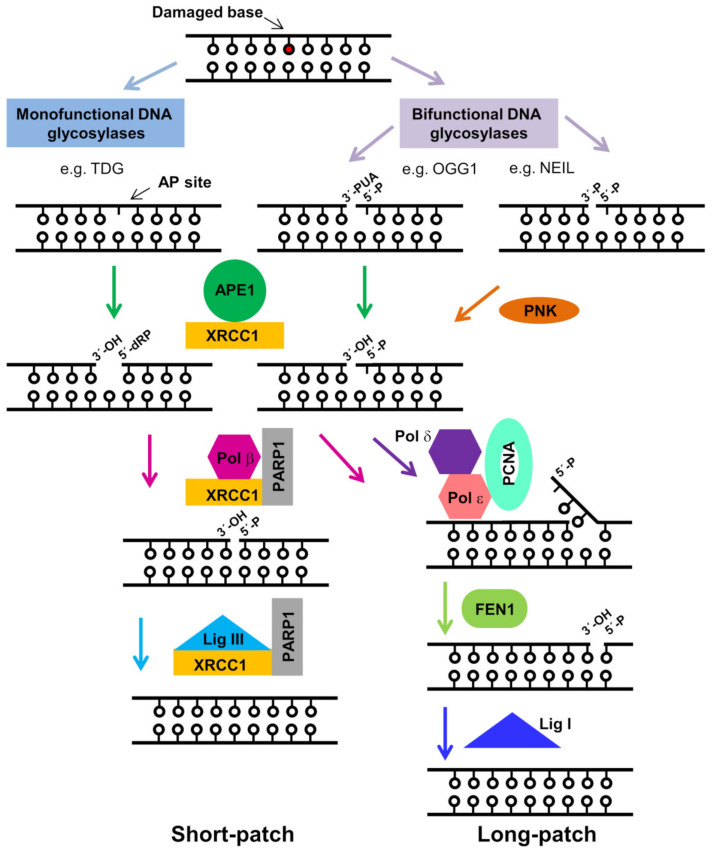
Scheme of short-patch (**left**) and long-patch (**right**) BER pathways. Depending on the type of base lesion (red circle), specific DNA glycosylases operate, leaving different terminal chemical groups. In addition to its polymerase activity, Pol β affords 5′-dRP lyase catalysis. Several of the BER enzymes rely on the scaffolding protein XRCC1.

**Figure 4 ijms-22-06308-f004:**
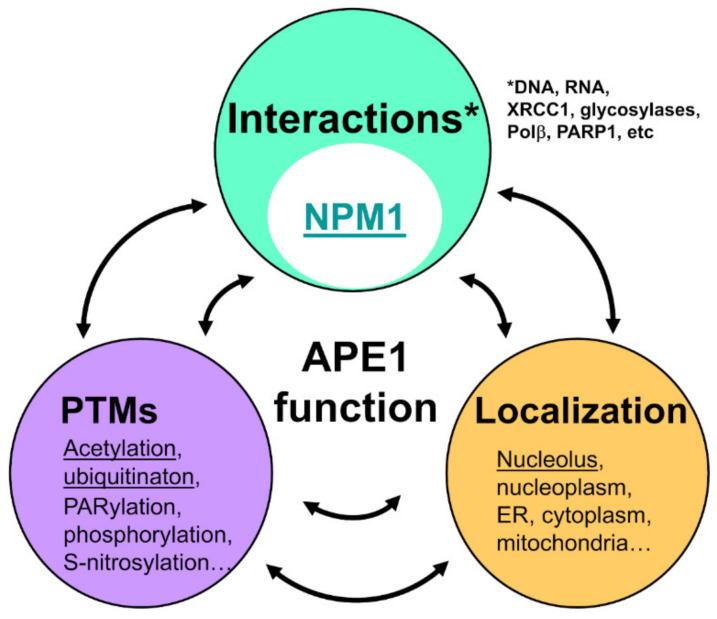
Interplay between the three main factors operating on APE1 regulation. Crosstalk among subcellular localization (mainly nucleolar), post-translational modifications (PTMs, acetylation and ubiquitination particularly targeting the N-terminal tail), and protein interactions in regulation of APE1. Regulation by NPM1, discussed in the text, further operates through control of localization, and is affected by APE1 acetylation state.

**Figure 5 ijms-22-06308-f005:**
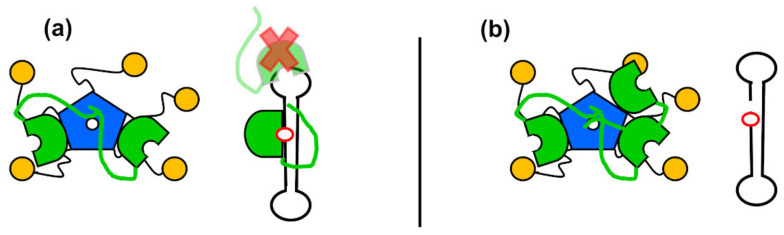
Hypothetical role of NPM1 in functional regulation of APE1. (**a**) Pentameric NPM1 (with the core domain depicted in blue and the C-terminal domains in yellow) can bind several molecules of APE1 (green) and compete with the unspecific, low affinity binding of APE1 to a model of substrate DNA, thus favouring the specific binding of APE1 to the abasic site (red circle); (**b**) after incision, turn-over of APE1 could be facilitated by NPM1, which would act as a chaperone reservoir.

## Data Availability

Not applicable.

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
