# Peer review of "Molecular Mechanisms Regulating the DNA Repair Protein APE1: A Focus on Its Flexible N-Terminal Tail Domain"

_ijms, 2021, doi:10.3390/ijms22126308_

Round 1

Reviewer 1 Report

This this manuscript titled “Molecular mechanisms regulating the DNA repair protein APE1: a focus on its flexible N-terminal tail domain”, the authors provide a historical and comprehensive perspective on the function and mechanisms of a critical repair protein APE1 especially its NTD in DNA repair. This is an important and critical topic in the field of genome integrity. The readers will be benefit from reading this perspective.

The manuscript is certainly well written and organized. However, below concerns (major and minor) should be address in a revised manuscript before acceptance for publication.

Major concerns:

  1. In the Section 2 “The role of APE1 in the BER pathway” on Page 2-3, the authors also need to briefly describe long-patch BER pathway in addition to short-patch BER pathway. In addition, the authors need to summarize the general description of the BER pathway, and point out the updated understanding on APE1’s detailed 3' - phosphodiesterase and 3' to 5'- exonuclease activities in BER and NIR pathway and corresponding key residues for its nuclease activities.
  2. In the Section of 3 “Role of APE1 beyond DNA repair” on page 4, the authors mentioned redox transcription, RNA metabolism, and DNA regulatory processes. However, the recently characterized role of APE1 in the ATR-Chk1 DDR pathway should also be included and described (Lin et al., NAR, 2020).
  3. In the Section 6.1 “The N-terminal tail and APE1 subcellular localization” on Page 8, the authors focused on the subcellular localization of APE1. Probably it is more accurate to describe extracellular localization of APE1. Several studies have demonstrated that APE1 can be secreted into plasma or cell culture, and involved in anti-inflammatory activities and cell apoptosis as a paracrine molecule in triggering cell-to-cell communication (Nath et al., Cell Signal, 2017; Park et al., Sci Reports, 2016; Pascut et al., Oncotarget, 2019). A recent study has demonstrated Ape1 is secreted by cells through the exosomes (Mangiapane et al., JBC, 2021). When we talk about APE1 subcellular localization, mitochondria are undoubtedly worthy to discussed deeply. More and more papers relating APE1 and mitochondria function occurred in past year, regarding mtDNA repair and mitochondrial oxidative stress (Barchiesi et al., JMB, 2020; Bazzani et al., BMC Cancer, 2020).
  4. In the Section 6.2 “The N-terminal tail and APE1 post-translational modifications” on Page 9, the authors summarized PTMs of APE1 NTD especially acetylation and ubiquitination. However, phosphorylation at Thr19 and Thr45 may also need to be included and discussed. What are the roles and functions of PTMs on APE1 NTD? Does acetylated APE1 change its capacity of binding to chromatin or DNA, or subcellular localization, or protein-protein interactions?
  5. In the Section 7 “Nucleophosmin as a regulator of APE1 via the N-terminal tail” on page 10-11, the authors focused on the regulation of APE1 by nucleophosmin (NPM1). However, such interaction and function of between NPM1 and APE1 has been deeply discussed in author recent review articles in DNA repair and BBA in 2020. Thus, this part is suggested to be abridged to certain extent at least overlapping parts.
  6. It is also benefiting reader by introducing APE1 inhibitors that may target APE1 NTD. Alternatively, APE1 inhibitors can be part of future studies in Section 8.
  7. What is the biomedical significance of APE1 NTD? Relevance to cancer etiology, biomarker, or cancer therapy targeting molecule?

Minor concerns:

Please pay attention to the typos and grammars. For example:

  1. In the second paragraph of Introduction in Page 2, authors stated “Altered expression levels … reported in several tumours, … being considered a relevant target….”. “several tumours” need to be changed by “several types of tumors”. “being considered a relevant target” need to be revised to “being considered as a relevant target”.
  2. In the last sentence of Introduction on Page 2, the authors stated “… this enzyme are fine-tuned”. ” … this enzyme are fine-tuned” need to be corrected by ” … this enzyme is fine-tuned”.
  3. In the first sentence of Page 3 “…DNA ligase III, further co-ordinated by…”, “co-ordinated” needs to be replaced by “coordinated”.

Author Response

Major concerns:

  1. In the Section 2 “The role of APE1 in the BER pathway” on Page 2-3, the authors also need to briefly describe long-patch BER pathway in addition to short-patch BER pathway. In addition, the authors need to summarize the general description of the BER pathway, and point out the updated understanding on APE1’s detailed 3' - phosphodiesterase and 3' to 5'- exonuclease activities in BER and NIR pathway and corresponding key residues for its nuclease activities.

We have described in more detail long-patch BER (at the end of the third paragraph of section 2) and included this sub-pathway in Figure 1.  We have removed some parts of the BER description, to summarize. We have better described the 3´-end processing activities (second paragraph), and mentioned additional key residues for catalysis in Section 4.

  1. In the Section of 3 “Role of APE1 beyond DNA repair” on page 4, the authors mentioned redox transcription, RNA metabolism, and DNA regulatory processes. However, the recently characterized role of APE1 in the ATR-Chk1 DDR pathway should also be included and described (Lin et al., NAR, 2020).

This APE1 role and the article describing it are now mentioned at the end of section 2 (better than section 3, which is dedicated to roles “beyond DNA repair”).

  1. In the Section 6.1 “The N-terminal tail and APE1 subcellular localization” on Page 8, the authors focused on the subcellular localization of APE1. Probably it is more accurate to describe extracellular localization of APE1. Several studies have demonstrated that APE1 can be secreted into plasma or cell culture, and involved in anti-inflammatory activities and cell apoptosis as a paracrine molecule in triggering cell-to-cell communication (Nath et al., Cell Signal, 2017; Park et al., Sci Reports, 2016; Pascut et al., Oncotarget, 2019). A recent study has demonstrated Ape1 is secreted by cells through the exosomes (Mangiapane et al., JBC, 2021). When we talk about APE1 subcellular localization, mitochondria are undoubtedly worthy to discussed deeply. More and more papers relating APE1 and mitochondria function occurred in past year, regarding mtDNA repair and mitochondrial oxidative stress (Barchiesi et al., JMB, 2020; Bazzani et al., BMC Cancer, 2020).

We agree that secretion is an interesting novel aspect of APE1 and have included a discussion on these findings. We are also mentioning the recent data on mitochondrial translocation, citing several of the suggested references.

  1. In the Section 6.2 “The N-terminal tail and APE1 post-translational modifications” on Page 9, the authors summarized PTMs of APE1 NTD especially acetylation and ubiquitination. However, phosphorylation at Thr19 and Thr45 may also need to be included and discussed. What are the roles and functions of PTMs on APE1 NTD? Does acetylated APE1 change its capacity of binding to chromatin or DNA, or subcellular localization, or protein-protein interactions?

Phosphorylation at those residues is now mentioned in section 6.2 and reflected in Figure 3. The crosstalk between PTMs, subcellular localization and protein-protein interactions is something we are trying to highlight and is now emphasized at the beginning of Section 6.

  1. In the Section 7 “Nucleophosmin as a regulator of APE1 via the N-terminal tail” on page 10-11, the authors focused on the regulation of APE1 by nucleophosmin (NPM1). However, such interaction and function of between NPM1 and APE1 has been deeply discussed in author recent review articles in DNA repair and BBA in 2020. Thus, this part is suggested to be abridged to certain extent at least overlapping parts.

This part has been shortened, removing parts of the second, third and fourth paragraphs.

  1. It is also benefiting reader by introducing APE1 inhibitors that may target APE1 NTD. Alternatively, APE1 inhibitors can be part of future studies in Section 8.

A new, brief section 8 on APE1 pharmacological targeting has been included, where the possibility of targeting the NTD-mediated interaction with NPM1 is mentioned.

  1. What is the biomedical significance of APE1 NTD? Relevance to cancer etiology, biomarker, or cancer therapy targeting molecule?

Please see the previous answer.

Minor concerns:

Please pay attention to the typos and grammars. For example:

  1. In the second paragraph of Introduction in Page 2, authors stated “Altered expression levels … reported in several tumours, … being considered a relevant target….”. “several tumours” need to be changed by “several types of tumors”. “being considered a relevant target” need to be revised to “being considered as a relevant target”.

These sentences have been corrected.

  1. In the last sentence of Introduction on Page 2, the authors stated “… this enzyme are fine-tuned”. ” … this enzyme are fine-tuned” need to be corrected by ” … this enzyme is fine-tuned”.

This was a plural (“how the multifaceted functions of this enzyme are fine-tuned”), but anyway the sentence has been rewritten for the sake of clarity.

  1. In the first sentence of Page 3 “…DNA ligase III, further co-ordinated by…”, “co-ordinated” needs to be replaced by “coordinated”.

It has been corrected.

Reviewer 2 Report

The article entitled “Molecular mechanisms regulating the DNA repair protein APE1: a focus on its flexible N-terminal tail domain” was taken under consideration the APE1 protein.

The topis of the manuscript is important for the broad scientific community in the DNA damage and repair fields, therefore it can be a valuable paper. Moreover, it is well written and readable.

However, some correction should be performed before publication:

In the article it is APE1 (apurinic/apyrimidinic endonuclease 1), in fact, it should be DNA apurinic/apyrimidinic site endonuclease 1 for better protein activity description.

The non-bulky lesion like AT inter-strand crosslink, cyclopurines are not substrates for BER machinery. Follow Chatgilialoglu cyclopurines are frequent lesions (Cells doi.:10.3390/cells8111303).

The influence of clustered and tandem DNA damage/lesion on APE1 activity should be described due to its importance for radio/chemo-therapy. Therefore, the work of Karwowski should be mentioned (doi:10.3390/cells8111303)

It would be a wonder if authors will provide some more data in the fields of why APE1 is a potential pharmacological target.

The figure described APE1 activity from the chemical point of view is highly required.

The authors discussed LP-BER but Figure 1 has shown only short patch BER, it must be corrected.

I think that the part about RNA and APR1 “interaction” should be removed from the manuscript. It is a broad topic for the next review article.

Instead of the above authors should put their efforts to describe in detail the protein “communication” by electron transfer (red-ox process).

In Figure 2 C-termini of the proteins should be indicated as well as Figure 2 should be more readable (maybe different colors should be used)

Authors used in the present article the wrong description of  pol b, 3’-a,b-unsaturated, etc. – the Greek letters are highly required.

Author Response

In the article it is APE1 (apurinic/apyrimidinic endonuclease 1), in fact, it should be DNA apurinic/apyrimidinic site endonuclease 1 for better protein activity description.

The name of the protein has been changed as suggested.

The non-bulky lesion like AT inter-strand crosslink, cyclopurines are not substrates for BER machinery. Follow Chatgilialoglu cyclopurines are frequent lesions (Cells doi.:10.3390/cells8111303).

These exceptions are now mentioned in the beginning of section 2 and in new section 8, and the reference cited.

The influence of clustered and tandem DNA damage/lesion on APE1 activity should be described due to its importance for radio/chemo-therapy. Therefore, the work of Karwowski should be mentioned (doi:10.3390/cells8111303).

This is now mentioned in Section 2 and in a new section 8 on APE1 pharmacological targeting, and the reference cited.

It would be a wonder if authors will provide some more data in the fields of why APE1 is a potential pharmacological target.

A new, brief section 8 on APE1 pharmacological targeting has been included.

The figure described APE1 activity from the chemical point of view is highly required.

A panel describing APE1 endonucleolytic reaction has been included in Figure 2.

The authors discussed LP-BER but Figure 1 has shown only short patch BER, it must be corrected.

We have included the LP-BER pathway in new Figure 1.

I think that the part about RNA and APR1 “interaction” should be removed from the manuscript. It is a broad topic for the next review article.

This section has been shortened, but since it is indeed a broad and important topic, where the N-terminal tail, focus of our review, has an important role, we feel that it needs to be briefly discussed.

Instead of the above authors should put their efforts to describe in detail the protein “communication” by electron transfer (red-ox process).

We have described in deeper detail the redox function, in the second paragraph of section 3.

In Figure 2 C-termini of the proteins should be indicated as well as Figure 2 should be more readable (maybe different colors should be used).

The C-termini are now indicated. We are using three colours in this figure, but they are not so evident by the high overlap between the three structures. In fact that is the point highlighted by the figure.

Authors used in the present article the wrong description of  pol b, 3’-a,b-unsaturated, etc. – the Greek letters are highly required.

We have checked the use of greek characters where necessary.

Round 2

Reviewer 1 Report

None

Author Response

Thank you

Reviewer 2 Report

The authors made some efforts; however, I have some critical remarks.

The new version of the article is extremely hard to read due to the old part's presence.

Therefore, the text needs to be once again checked toward typing and grama mistakes.

Authors have still forgotten the Greek letters.

The publication (doi:10.3390/cells8111303) bout hAPE1 and cyclo-2’deoxypuries must be cited, as I have mentioned previously.

Author Response

Please see ALSO the attachment.

COMMENT: The authors made some efforts; however, I have some critical remarks.

The new version of the article is extremely hard to read due to the old part's presence.

REPLY: The “old parts” need to be present in the word document as required by the Editorial Office.

Please read the final, corrected version of the manuscript, where the “old parts” are not displayed. You can find it in the pdf file attached, where the corrected parts are highlighted in yellow (1st revision) and green (2nd revision).

COMMENT: Therefore, the text needs to be once again checked toward typing and grama mistakes.

REPLY: Please see the final pdf version, which we have thoroughly checked for typos and grammatical mistakes. Still, in case you find something wrong, we would much appreciate if you could please point it out to us the specific mistake, and we will of course correct it.

COMMENT: Authors have still forgotten the Greek letters.

REPLY: Hopefully, you will find the Greek characters properly transcribed in the pdf version.

COMMENT: The publication (doi:10.3390/cells8111303) bout hAPE1 and cyclo-2’deoxypuries must be cited, as I have mentioned previously.

REPLY: The suggested publication is now cited in a new sentence in Section 8.
